# Parameters Influencing Moisture Diffusion in Epoxy-Based Materials during Hygrothermal Ageing—A Review by Statistical Analysis

**DOI:** 10.3390/polym14142832

**Published:** 2022-07-12

**Authors:** Camille Gillet, Ferhat Tamssaouet, Bouchra Hassoune-Rhabbour, Tatiana Tchalla, Valérie Nassiet

**Affiliations:** 1Laboratoire Génie de Production, INP-ENIT, Université de Toulouse, 47 Av. d’Azereix, 65000 Tarbes, France; bouchra.hassoune-rhabbour@enit.fr; 2Safran Aircraft Engines, site de Villaroche, 77550 Moissy-Cramayel, France; tatiana.tchalla@safrangroup.com; 3Laboratoire PROMES-CNRS (UPR 8521), Université de Perpignan, TECNOSUD, 66100 Perpignan, France; ferhat.tamssaouet@univ-perp.fr

**Keywords:** epoxy resin, composite material, hygrothermal ageing, water diffusion, Fick model deviation, statistical analysis, box plot, PCA

## Abstract

The hygrothermal ageing of epoxy resins and epoxy matrix composite materials has been studied many times in the literature. Models have been developed to represent the diffusion behaviour of the materials. For reversible diffusions, Fick, Dual–Fick and Carter and Kibler models are widely used. Many parameters, correlated or not, have been identified. The objectives of this review by statistical analysis are to confirm or infirm these correlations, to highlight other correlations if they exist, and to establish which are the most important to study. This study focuses on the parameters of the Fick, Dual–Fick and Carter and Kibler models. For this purpose, statistical analyses are performed on data extracted and calculated from individuals described in the literature. Box plot and PCA analyses were chosen. Differences are then noticeable according to the different qualitative parameters chosen in the study. Moreover, correlations, already observed in the literature for quantitative variables, are confirmed. On the other hand, differences appear which may suggest that the models used are inappropriate for certain materials.

## 1. Introduction

For many decades, epoxy-matrix composites have been widely used in many industries, especially in aeronautics. Airframers seek to lighten the onboard aircraft mass as much as possible to reduce their fossil energy consumption. From that perspective, metals are substituted by polymer-matrix composite materials, which offer very high mechanical properties for a significantly lower density. Epoxy is a commonly used thermosetting polymer family, whether as a matrix in composite materials or as a structural adhesive for bonded joints and repairs in the aeronautical industry. Epoxy prepolymers and their hardeners form non-cross-linked systems with low viscosity, which ease their processing. They cover a wide range of cross-linking temperatures (from 5 to over 200 °C), depending on their chemical composition and the intended utilization. The epoxy system shrinkage after cross-linking is low compared to other thermosetting materials such as phenolic resins, making them good candidates as adhesives. In addition, the presence of polar functional groups such as hydroxyl gives them the ability to adhere to substrates or fibres. Besides, epoxies are commonly used as a matrix for composite materials because of their good mechanical properties and chemical resistance to many corrosive substances and acids. Epoxies also have good dielectric properties, allowing them to be electrical insulators [1,2].

However, despite their many beneficial properties, epoxy-based materials are sensitive to wet exposure as water molecules can penetrate their macromolecular networks. Water molecule action can be reversible, as in the case of plasticization, or irreversible, such as hydrolysis. In the first case, the water molecules break the secondary bonds between neighbouring chains and partially destroy the mechanical cohesion of the polymer, which results in: (1) a decrease in the glass transition temperature $T_{g}$, which can be reversible; as well as (2) a loss of additives and the material swelling [3,4,5]. These swellings cause concentrations of mechanical stresses that can eventually lead to irreversible damage, such as decohesion between the fibre and the matrix or microcracks. Due to the difference in elasticity between the fibres and the matrix, and the water absorption, stresses develop along with the fibre/matrix interface. Within the sample, stress equilibrium is more easily maintained. However, at the surface, edge stresses are high enough to produce cracks. After cyclic exposure to a moist environment, micro-cracks and disbonds at the fibre/matrix interface tend to coalesce and expand [6,7,8,9,10]. In the case of hydrolysis, the water molecules penetrate directly to the macromolecule skeleton causing chain breaks, which destroy the cohesion of the material and allow the formation and propagation of cracks [11,12]. This decrease in mechanical properties is greatly impacting and has been widely demonstrated in the literature [13,14,15,16,17,18,19,20,21].

The literature has proved several times that hygrothermal ageing is governed by many parameters. These latter include, firstly: material parameters such as prepolymer type [22,23,24,25], hardener type [22,23,26], reinforcement type [13,27,28,29] or thickness [30,31]; secondly: conditioning parameters such as temperature [4,6,32] and relative humidity [22,33,34]; and thirdly: the resulting diffusion parameters such as saturation water mass uptake, saturation time and diffusivity [35]. Some of these parameters are correlated, meaning that they evolve together, while others are independent. The objective of this paper is to explore how these parameters influence each other and determine the principles behind this. Comparisons of water mass uptake by epoxy resin and epoxy-based composites have already been made in several publications, but without considering the variation of all the parameters [24]. A classical approach to comparing the different ageing mechanisms does not seem to be sufficient for a large dataset, because of the heterogeneity of the results.

This paper aims to confirm the correlations found by the literature between the different variables influencing the hygrothermal ageing and emphasize less apparent correlations. The objective is to highlight the most important variables of the problem, i.e., the most important to study.

Therefore, statistical studies on a large number of data were carried out. They include information on material type, ageing conditioning and parameters describing the diffusion curve. Data were extracted and calculated from *90 publications* on hygrothermal ageing of epoxy and epoxy-based composites: [4,5,6,7,10,13,15,16,18,19,22,24,26,27,28,29,30,31,32,33,34,36,37,38,39,40,41,42,43,44,45,46,47,48,49,50,51,52,53,54,55,56,57,58,59,60,61,62,63,64,65,66,67,68,69,70,71,72,73,74,75,76,77,78,79,80,81,82,83,84,85,86,87,88,89,90,91,92,93,94,95,96,97,98,99,100,101,102,103,104]. This study, which includes *448 individuals*, focuses on reversible gravimetric moisture uptake curves exhibiting Fick or Fick-derived two-step diffusion behaviour; water uptake is a function of the root of time. Irreversible behaviours, in particular those with mass losses during wet ageing or high mass uptake without asymptote, have been excluded from this study in order to avoid bringing together too different phenomena [22,32,105].

First, the diffusion parameters and the models to which they are related are presented. Then, the dataset is represented in order to study the relationships between the variables. For this purpose, box plots and scatter plots are generated to observe the variables’ dispersion. However, the wide dispersion of the data could make it difficult to draw conclusions by simply observing the evolution of the variables between them. These data could present a great heterogeneity caused by the great diversity of the studied materials due to their different nature and process and the different conditioning parameters. All these parameters vary at the same time from one individual to another, which leads to different absorption behaviours, knowing that irreversible behaviours have already been ruled out. To complete the statistical study, the principal component analysis (PCA) is used. This analysis is effective for deducing correlations when a large number of quantitative variables is involved. By reducing and centering the variables, PCA facilitates observations despite their large dispersion. This includes:Highlighting material and ageing parameters influencing the diffusion parameters, by box plots and scatter plots;The correlation of the different quantitative parameters of the study by PCA;The separation of the data into four classified PCAs in order to propose more efficient correlations;Evidence of differences between PCA depending on the diffusion model and the type of material (resin or composite);Discussion of the limitations of using the diffusion models on epoxy-based materials.

## 2. Studied Parameters

The determination of diffusion parameters in the case of composites has been extensively study in the literature, with respect to the material configuration, the diffusion duration, and/or the diffusivity linearity, etc. [35,37,45,65,106,107,108].

Various models have been developed to describe the reversible behaviour of water absorption. The most well-known is the Fick model, for diffusion behaviour without anomaly, with a linear slope followed by saturation [109]. However, many epoxies show behaviours that deviates from the Fick model with the presence of two diffusion steps. Various models have been proposed to consider these anomalies, including the Dual–Fick model [20,110,111] and the Carter and Kibler model [19,21,45,51,112,113]. The equations of these three models are given below: Fick (Equation (Equation 1)), Dual–Fick (Equation (Equation 2)) and Carter and Kibler (Equation (Equation 3)).
(1)$$M\left(t\right)=M_{sat}\left(1-\frac{8}{\pi^{2}}\sum_{n=0}^{\infty }\frac{1}{{\left(2n+1\right)}^{2}}\cdot exp\left[-{\left(2n+1\right)}^{2}\cdot \pi^{2}\cdot \frac{D\cdot t}{h^{2}}\right]\right)$$
(2)$$M\left(t\right)=\sum_{i=1}^{2}\left[M_{sat_{i}}\left(1-\frac{8}{\pi^{2}}\sum_{n=0}^{\infty }\frac{1}{{\left(2n+1\right)}^{2}}\cdot exp\left[-{\left(2n+1\right)}^{2}\cdot \pi^{2}\cdot \frac{D_{i}\cdot t}{h^{2}}\right]\right)\right]$$
(3)$$M\left(t\right)=M_{sat}\left(1-\frac{8}{\pi^{2}}\sum_{l=1}^{\infty }\frac{r_{l}^{+}exp\left(-r_{l}^{-}t\right)-r_{l}^{-}exp\left(-r_{l}^{+}t\right)}{l^{2}\left(r_{l}^{+}-r_{l}^{-}\right)}+\frac{8}{\pi^{2}}\frac{\kappa \beta }{l(\gamma +\beta )}\sum_{l=1}^{\infty }\frac{exp\left(-r_{l}^{-}t\right)-exp\left(-r_{l}^{+}t\right)}{r_{l}^{+}-r_{l}^{-}}\right).$$

In this study, the different parameters defining the diffusion are taken from those three models. For Fick behaviour individuals, the parameters selected are the saturation water mass uptake $M_{sat}$, the saturation time $t_{sat}$ and the diffusivity *D*. Additional parameters are included to refine the observations for the two-step diffusion behaviour (abbreviated to Dual) individuals. Two parameters come from the Dual–Fick model: The water mass uptake at the first absorption step $M_{inter}$, and its time to be reached $t_{inter}$ (Figure 1). The last three parameters, determined using the model developed by Carter and Kibler, that takes into account the functional groups of hydrophilicity, are: (1) γ the probability per time unit that free water molecules in the polymer will bond; (2) β the probability per time unit that bonded water molecules will liberate; and (3) *K* a parameter related to the material swelling. For the homogeneity purpose, all these parameters have been recalculated using the Carter and Kibler approximations, as a reminder [45]:

1- At the first step (intersection point between the two sorption stages):(4)$$\frac{M_{inter}}{M_{sat}}≅\frac{\beta }{\gamma +\beta };$$

2- At short times (curve linear domain):(5)$$\frac{M_{inter}}{M_{sat}}=\frac{4}{\pi^{3/2}}\left(\frac{\beta }{\gamma +\beta }\right)\sqrt{Kt};$$

3- At longer times (beyond the first step):(6)$$\frac{M(t)}{M_{sat}}≅\left\{1-\frac{\gamma }{\gamma +\beta }\left(e^{-\beta t}\right)\right\};$$
where: $K=\frac{\pi^{2}D}{2h^{2}}$, *D* is the diffusivity, and *h* is the sample thickness.

All these parameters are related to the material parameters: prepolymer type, hardener type, reinforcement type, fibre architecture, fibre volume fraction $v_{f}$, thickness *h*, process; and conditioning parameters: conditioning (immersion in water or exposure to humid air), relative humidity RH, and ageing temperature $T_{ageing}$. Some variables are qualitative while others are quantitative (Table 1).

## 3. Descriptive Statistics: Study of Dispersion

To observe the variables’ dispersion and study their mutual influence, the whole dataset is represented using box plots. This allows the removal of outliers and the highlighting of the median and central quartile values. As a result, the dataset is separated into four equally sized portions. Each of the three values that divide the elements of a statistical distribution is called a quartile. The box contains 50% of the individuals: 25% between the first quartile and the second quartile-or median-and 25% between the median and the third quartile. The “whiskers” are the lines that run across the box. They indicate the variability outside the quartiles, and each includes 25% of the individuals. The points outside the whiskers are atypical values.

With this first scatter analysis, the entire dataset is observed. The dispersion of the quantitative values saturation mass uptake $M_{sat}$, diffusivity *D* and saturation time scaled to thickness $t_{sat}/h$ are studied, as a function of the categorical variables that describe the dataset: diffusion behaviour, prepolymer type, hardener type, reinforcement type, fibre architecture, process, and conditioning (Table 1). Time was reduced to thickness to normalize the results and obtain comparable durations. These three quantitative variables were also observed as a function of other quantitative variables: fibre volume fraction, relative humidity rate RH, and ageing temperature $T_{ageing}$. For this purpose, scatter plots were generated.

The *R* programming language and software, its packages *corrplot* and *factoextra*, as well as the *boxplot* function are used [114,115,116].

### 3.1. Diffusion Behaviour

The two types of moisture diffusion behaviour, Fick and Dual diffusions, are studied. Figure 2 shows the data distribution within quartiles for the variables $t_{sat}/h$, *D* and $M_{sat}$. All characteristic values of the box plots are shown. Outliers are not shown for the sake of clarity, but their percentage is indicated. One can observe that the distributions are not centered, and all the variables and behaviours have a higher skewness, which shows that they are not normal (Gaussian). This is reflected in the fact that some individuals have higher mass absorption, longer saturation time, and greater diffusivity, i.e., greater sensitivity to moisture. The box plot shows that the same median is obtained regardless of the moisture diffusion behaviour. In contrast, the third quartile and the upper whisker boundary are larger for specimens with two diffusion steps. As many individuals are located on either side of the median, but beyond this median, they are more widely spaced, and some have a large diffusivity that stretches the graph. Diffusivity and saturation time reduced to thickness reveal a very wide dispersion from one behaviour to another. For $t_{sat}/h$, the gap between the two behaviour types is around $10^{2}$ for the median, which is very large.

For the saturation mass uptake, the median is slightly lower for Fick individuals than for individuals with two diffusion steps. For Fick individuals, the distance between quartiles is similar. A low dispersion of epoxies and epoxy-based composites is observed. Indeed, in this study, 75% of the individuals have a moisture absorption lower than 4%. The diffusivity and the saturation time show strong dispersion. While saturation can take considerable time, the saturation mass uptake varies only slightly after passing the first diffusion step. These differences in diffusion behaviour depend on the different parameters given in the Table 1. Indeed, the diffusion parameters differ according to the material parameters and the ageing parameters, detailed in the following.

### 3.2. Epoxy Prepolymer Type

Some polymers are more likely to absorb water due to their hydrophilicity [102,117,118,119], or the free volume presence [120,121,122]. Among the epoxy family, differences were found in the prepolymers functional groups and in their structure. The dataset is sorted according to the prepolymer type in order to observe the variation in the dispersion of $M_{sat}$, $t_{sat}/h$ and *D*. For the sake of simplicity, only medians are shown (Table 2) and the entire box plots are available as Appendix A. Medians divide the studied populations into two sets comprising the same number of individuals. One can notice that these individuals show high skewness similarly to the box plots based on diffusion behaviour. Looking at the saturation mass uptake, we find values about 1.05% for the DGEBF resin and 6.37% for the DGEBA + novolac resin. In contrast, the DGEBA and novolac resins have weaker saturation mass uptake when unblended. The other blends (DGEBA + mTGAP, DGEBA + TGDDM) also appear to have lower moisture absorption than the neat resins. Furthermore, the mTGAP resins also show strong moisture absorption, which can be explained by the fact that they are two aromatic epoxy prepolymers with three oxirane groups and a nitrogen atom, which are very hydrophilic [25,117,118,123,124].

Indeed, the hydrophilicity of the polymer is linked to the nature of the chemical groups composing the macromolecules. Water molecules are attracted to polar functions, such as hydroxyls or nitrogens. According to Van Krevelen, water absorption is an additive molar function. For a representative structural unit, independent of its environment, it is written (Equation (Equation 7)) [117,118]:(7)$$H=\frac{M(t)\cdot M}{1800},$$
where *H* is the number of water moles, *M* is the molar mass and M(t) is the percentage of mass uptake. There are universal values of *H*. For non-polar hydrocarbons (–CH–, –CH${}_{2}$–, –CH${}_{3}$), fluorinated groups or aromatic rings: H=0. Moderately polar groups, such as esters and ethers, have H<0.3. Hydrogen bonding groups, e.g., acids, alcohols, amides, amines and hydroxyls, are very hydrophilic and have: H=1–2 [25,118,123,124,125]. Some structures combine hydrophilic and hydrophobic groups, such as polyamides and epoxies. The hydrophilicity then depends on the respective proportions of these groups. These predictions apply in the case where the contributions of the different groups are independent. For epoxies, where internal hydrogen bonds can interact with hydrogen bonds caused by moisture uptake, it is necessary to consider structural units that include groups interacting with each other [101,102,119,126].

However, the TGDDM prepolymers, composed of four oxirane groups and two nitrogen atoms, do not experience such large moisture absorption. Another phenomenon that affects moisture uptake is the gelation rate. Indeed, faster gelling traps more voids and has a lower cross-linking density. For example, Frank et al. report this to be the case for mTGAP/DDS, which has a greater saturation mass uptake than DGEBA/DDS. Nevertheless, the size of the free volumes decreases with the macromolecule functionality because of the increase of the cross-linking density. So although TGDDM/DDS has four functionalities against three for mTGAP/DDS, its longer and more flexible skeleton allows it to have smaller free volumes and lower absorption [41,44,119].

Different profiles are observed in Table 2. The highest absorption rates are not always associated with large diffusivities or long saturation times. The more polar functions the epoxy has, the greater the diffusion temperature dependence and the more hydrophilic it is. In this study, a pretty low saturation mass uptake is associated with a high saturation time and a low diffusivity, and vice versa: for example, DGEBF, ESO, TGDDM or DGEBA resins. On the other hand, not all specimens respect this behaviour as the DGEBA + Novolac mixtures and mTGAP combine low diffusivities, long saturation time and high saturation mass uptake. The diffusion depends on the absorption curve shape. According to [25], polar sites in hydrophilic systems could act as a barrier to diffusion. Polar groups act as “bottlenecks” and trap incoming water molecules, named bonded water, which inhibits their free diffusion within the free volumes. Considering the data dispersion, the curing effect on moisture sensitivity should also be taken into account. Indeed, Abdelkader and White find that imperfect resin cross-linking leaves uncrosslinked epoxy areas that are more sensitive to water and enable faster penetration into the material. In addition, cross-linking at excessive temperatures could reduce the polymer density through the porosities formation. These porosities allow water molecules to have more free volumes in which to settle [26,102].

### 3.3. Hardener Type

Due to the large number of hardener types, they are classified according to their chemistry: amine, amidoamine, dicyandiamide, anhydrid acid and phenol novolac. Since 276 individuals are amine, or 62%, this category of hardeners was divided into four, according to their structure: aliphatic, cycloaliphatic, aromatic and unknown when not given in the publication (Table 3).

The classification by the type of hardener in this paper shows a lower dispersion than the classification by the type of prepolymer, for example, medians of the saturation mass uptake range from 0.79% to 3.20%. It should be taken into account that the crosslinking sites of thermoset materials are generally considered hydrophilic due to the high presence of hydrogen bridges, which tend to absorb more water [127,128,129]. For example, polyvinyl alcohol, polyacrylamides, amine and amide hardeners are very hydrophilic, which may explain this lower dispersion.

In the box plots, both aliphatic and aromatic amines show similar saturation mass uptake and saturation time medians. The diffusivity is lower for the aromatics. Cycloaliphatic amines have slightly lower saturation mass uptake, significantly lower diffusivity medians and longer saturation times. The unknown amines have much lower $M_{sat}$ and $t_{sat}/h$, but *D* close to those of aliphatic amines. This result conflicts with some observations in the literature, where aliphatic amines have higher saturation masses than aromatic amines [26,41]. Amidoamines have much lower $M_{sat}$ and $t_{sat}/h$ than aliphatic, aromatic and cycloaliphatic amines. On the other hand, they show the highest *D* medians of the study. The dicyandiamides have the highest $t_{sat}/h$ medians. The anhydrid acids have low $M_{sat}$ and $t_{sat}/h$. Phenol novolacs exhibit low $t_{sat}/h$, with $M_{sat}$ and *D* close to those of aromatic amines.

Hardeners with a bulky structure, such as amidoamines, enable us to obtain a flexible network because of their long macromolecules with a high functionality. The steric hindrance is very important, which causes significant free volume, allowing the rapid entrance of water molecules [26]. On the other hand, their chemistry does not present more hydrophilic sites than other types of hardeners, such as amines, which can explain the rather low $M_{sat}$ medians. For aromatic amines, such as dianilines, the variation in their sensitivity to moisture is related to their reactivity and polarity. Furthermore, the hydroxyl and tertiary amine groups play a concerted role in the water bonding to the network, making their group contributions indistinguishable. The formation of a water-amine hydrogen bond competes directly or indirectly with an internal hydroxylamine hydrogen bond whose strength increases with the amine nucleophilicity. The indirect effect would occur if the water placement required a site of a particular steric configuration which, in its turn, depends on the amine-hydroxyl interaction [26,119].

Among the cycloaliphatic amines, many are IPDA, which are quite hydrophobic due to their CH${}_{3}$ groups. These results have already been observed in the literature. The aliphatic and aromatic amines are all moderately hydrophilic due to their chemical composition. The most important saturation $M_{sat}$ are linked to relatively low $t_{sat}/h$ compared to the others. This may also apply to anhydride acid. In this hardener family, we observe hydrophobic specimens due to the CH${}_{3}$ presence, which reduces their hydrophilicity. This may be the case with acetic anhydride. However, they are also more sensitive to humidity anhydrides, such as methylhexahydrophthalic anhydride (MHHPA). Epoxy anhydrides subjected to absorption followed by hydrolysis-related mass losses have been reported in the literature [22]. Hydrolysis-sensitive hardeners, such as some anhydrides or dicyandiamides, could then have lower mass uptakes.

### 3.4. Fibres Presence

While epoxy resin matrices are subject to wet ageing, carbon fibres can be considered impervious to moisture. Similarly, glass fibres are considered to have a low sensitivity to moisture over the material lifetime. It is important to note that despite their impermeability, these inorganic fibres impact moisture diffusion as they modify the behaviour of the polymer at the fibre-matrix interface. Differential swelling occurs and develops high stresses. Decohesion occurs if the adhesion is not sufficient. A void then appears at the interface, and the water uses it to propagate more rapidly through the material, like a moisture carrier [8,39,118]. In the study, composites based on carbon and glass fibres have the lowest median absorbed water mass uptake, with 1.39% for carbon and 0.80% for glass (Table 4). Their median diffusivities and saturation times are also the lowest among all the reinforcement types. As glass and carbon fibres are not very sensitive to water, $M_{sat}$ should be reduced to the resin mass fraction in order to take into consideration only the matrix part in the composites. We also note that the $M_{sat}$ and *D* box plots for the glass fibres are centred, i.e., the specimens are homogeneously distributed on either side of the medians, which is equal to the mean, despite the differences in resin type and conditioning. This low dispersion of values may be due to the diffusion inertia associated with the presence of fibres.

The box plots highlight the flax, hemp and regenerated cellulose fibre composites, which have median saturation masses of 9.82, 13.0 and 7.10%, respectively. Their median diffusivity is slightly higher than the other materials. Thus, for the hemp and regenerated cellulose fibre individuals, the median saturation times is also higher. It is recognized in the literature that these organic vegetal fibres are sensitive to water. This sensitivity depends on their chemical composition, in particular on their lignin and hemicellulose content [130]. In the composite, in addition to the matrix, the fibres also absorb moisture. Indeed, organic fibres are subject to swelling, which can cause matrix cracking and accelerated water diffusion, resulting in a higher mass of absorbed water than in the neat resin [3,9,89]. Surface treatments can then be used on organic fibres to reduce their hydrophilicity and the moisture absorption they induce. For example, potassium hydroxide and sodium hydroxide are used to reduce the ability to create hydrogen bonds between natural fibres and water. For cellulose fibres, they will remove open hydroxyl groups. Silane can also be used to stabilize the fibres and make them resistant to leaching by masking hydroxyl groups, by creating silanol, and reducing the number of porosities [131,132,133]. Aramid fibres are also organic fibres, composed of very hydrophilic amide bonds, but they do not seem to cause disproportionate moisture absorption, with a median saturation mass uptake of 3.54%.

A composite material can have different behaviours on which the diffusion properties depend. The fibres’ presence initially leads to a change in the flow path of water molecules, creating anisotropic diffusion at the macroscopic scale, which is the composite scale. Zhou and Lucas studied the dimensional changes that can be caused by the presence of fibres on carbon/epoxy composites immersed in distilled water at different temperatures. The diffusion along the fibre shows extreme stability, and no dimensional changes were measured. This stability is due to the carbon fibres high longitudinal stiffness. As they are impermeable to water absorption, the longitudinal dimension of the fibre is considered invariant. In width, the fibres hold and block the matrix, which prevents peeling. However, in the thickness, there are layers of neat epoxy that are not blocked by the fibres, which can peel under severe ageing. If saturation is reached, there is no longer any dimensional change [7]. The water diffusion in the composite material is strongly dependent on the fibre volume fraction $v_{f}$, and their arrangement, which affects both the gap size between two fibres and the diffusion path length through the matrix [35,110,112,134]. The effect of fibre volume fraction on moisture diffusion is complex, as the statistical study shows in Figure 3. $M_{sat}$, $t_{sat}/h$ and *D* they show an increase as a function of $v_{f}$ up to a certain threshold beyond which they fall. The fewer the gaps between the fibres, the shorter the diffusion paths through the matrix. As the fibre volume fraction increases, the diffusivity increases. However, if it is too important, the fibres are grouped and can touch each other. In this situation, the local diffusion is blocked and therefore is very low [33,112,134]. In Figure 3, the evolution of the variables seems to be in accordance with these conclusions.

Observing the reinforcement architectures used in the dataset, the composites composed of long fibre mats have a very important median absorbed mass uptake, which can be explained by the large voids created during its manufacture process (Table 5). This median absorbed mass uptake is quite similar for reinforcements arranged in one direction (UD), in two directions (2D), balanced, or in three dimensions (3D). It is slightly higher for UD and balanced reinforcements than for 3D. The median diffusivities are the greatest for the balanced and 3D composites, and the median saturation times are also the shortest for these two reinforcement configurations. The water molecules may have more directions to propagate, and and saturation is reached more rapidly. The literature is not uniform on this issue. Tang et al. found by modelling that their balanced woven fabrics diffused faster than their UD, particularly when the fibre waviness was greater [134]. For Yuan et al., the experimental data and modelling reveal a greater water mass absorbed in their 3D than in their UD. They also report a higher bound water proportion throughout the material [95]. On the other hand, Almudaihesh et al. observe that their UD composites absorb more water than their woven composites [135]. Finally, Wan et al. obtained a lower saturation mass uptake and diffusivity for their 3D than their UD. Their suggestion is that the moisture diffusion path is distorted in the 3D, which exerts a greater hindrance than for the UD [136].

### 3.5. Manufacturing Process

Many processes have been developed to manufacture polymer or organic matrix composite materials. The process choice depends on various criteria: production speed, cost, desired performance, size and shape, resin and fibres nature. The quality of the process affects its cross-linking rate and the porosity percentage. Manufacturing processes applying pressure, such as resin transfer moulding (RTM), thermopressing or prepreg curing in the autoclave, result in a high fibre volume fraction and low porosity [136,137,138]. With these processes, the median saturation mass uptakes, saturation times and diffusivities appear relatively low. The dispersion of these values is also small (Table 6).

Contact moulding and heating table processes show high median diffusivity and mass uptake. These processes have the common feature that they do not use pressure. They do not allow for high volume fractions of fibres. In addition, the fibres distribution and the resin content are not uniform, which can lead to voids formation. They show a noticeable data dispersion, probably because the dispersion and the quantity of porosities are variable, depending on the manufacturing parameters used.

Infusion, which has intermediate water absorption properties, also shows quite a large data dispersion. The vacuum bag low compaction is the cause of a greater void formation compared to other pressurized processes such as autoclave, which is responsible for greater diffusion and moisture absorption. Nevertheless, it is possible to improve the polymer quality by influencing its viscosity. A high viscosity may be responsible for a higher porosity. This porosity, therefore, varies according to the chosen resin type and cross-linking parameters, which may explain the wide dispersion of the data [139,140].

Concerning the unknown manufacturing process materials, it may be interesting to specify that these are generally aeronautical composites for which the information has not been revealed in the source publication. They may therefore be materials with high properties, which explains their low saturation mass uptakes and diffusivities.

### 3.6. Ageing Conditions

#### 3.6.1. Conditioning Environment

Wet ageing can occur at various locations: in water, whether distilled water, seawater or deionised water, or in the humid air. In the literature, these environments have been widely studied. In addition, deionised water and humid inert atmosphere are also analyzed. By examining the individuals immersed in water, a similar median saturation mass uptake appears for distilled and deionised water, while it is much lower for seawater. Although the box plots for distilled water are almost centred, indicating good results homogeneity, the box plots for deionised water are eccentric towards the top (Table 7). The lowest saturation mass uptakes are obtained under humid air, while the highest are obtained in immersion in distilled or deionised water. Mass uptakes in seawater is also low. The mineral presence in the water slows down the water molecule diffusion through the material and leads to lower saturation masses, although these differences are mainly perceived over long ageing times [12,47,64]. In contrast, diffusivity under humid air is much higher and allows saturation to be reached most quickly. It is followed by the diffusivity under distilled water, which is very close. Finally, the diffusivity in deionised water appears to be the lowest, despite the absence of minerals that could slow down the water molecule diffusion. Nevertheless, in the database, the number of individuals immersed in deionised water or seawater is very low compared to the number of individuals immersed in distilled water, which could be linked to this divergence of results. It is possible that other parameters have a greater influence on the values of $M_{sat}$, $t_{sat}/h$ and *D* than the water type.

#### 3.6.2. Relative Humidity

The relative humidity percentage influence on the absorbed water mass by epoxy-based materials has been demonstrated many times in the literature and is no longer in doubt [18,45,51,53,54,55]. This statistical study confirms that the higher the relative humidity, the greater the saturation mass uptake $M_{sat}$ (Figure 4). Immersion of this material type in distilled water further increases its absorbed water mass [22].

This dataset effectively confirms the positive correlation between the absorbed water mass uptake and the relative humidity. Henry’s law allows this correlation to be represented for humid air conditioning (Equation (Equation 8)) [20,124].
(8)$$C_{sat}=SP_{s},$$
where $C_{sat}$ is the saturation water concentration and $P_{s}$ is the water partial pressure, linked with RH (Equation (Equation 9)).
(9)$$RH\left(\%\right)=100\frac{P_{s}}{P_{sat}}.$$

Several versions of this law adapted to Dual sorption were subsequently developed: power law (Equation (Equation 10)) [37]; Langmuir (Equation (Equation 11)) [141]; Dual sorption (Equation (Equation 12)) [142]; and Flory–Huggins (Equation (Equation 13)) [143].
(10)$$C_{sat}=a{\left(\frac{P_{s}}{P_{sat}}\right)}^{b}$$
(11)$$C_{sat}=\frac{cP_{s}}{1+dP_{s}}$$
(12)$$C_{sat}=SP_{s}+\frac{cP_{s}}{1+dP_{s}}$$
(13)$$\ln a_{s}=\ln\frac{P}{P_{s}}=\ln v+\left(1-v\right)+\chi {\left(1-v\right)}^{2},$$
with *a*, *b*, *c* and *d* as coefficients, $a_{s}$ the solvent activity, *v* the volume, and χ the polymer-solvent interaction coefficient. For pure resins, *b* is between 1.3 and 1.8 while it is close to 1 for composite materials. *c* and *d* are given approximately by the statistical thermodynamic treatment of Langmuir’s.

In immersion in a liquid, $C_{sat}$ is related to the chemical potential of the water, i.e., it tends to decrease when the concentration of solutes increases [118]. The solvent concentration [S] in an environment is in equilibrium with its partial pressure $P_{s}$ in the atmosphere. There is a maximum concentration ${\left[S\right]}_{sat}$, which corresponds to the saturation pressure $P_{sat}$ (Equation (Equation 14)).
(14)$$\frac{P}{P_{s}}=\frac{\left[S\right]}{{\left[S\right]}_{sat}}.$$

If there is no extraction of soluble species, the polymer behaves in the same way in the liquid as in the saturated vapour. Indeed, equilibrium corresponds to the equality of the solvent chemical potentials in the polymer and in the environment. If the material is damaged, has pores or cracks, the solvent can also flow into it, as part of irreversible ageing. The presence of solutes in water, such as salt in seawater, causes the decrease of chemical potential, of saturation vapour pressure, and of solvent equilibrium concentration. Pure water causes greater moisture uptake than mineral water or seawater [20].

The diffusivity *D* also increases with the relative humidity in the environment. The calculation of this, according to the Fick model, is directly linked to $M_{sat}$, associated with the RH, and the thickness *h* (Equation (Equation 16)) [109,144]. For its part, the Carter and Kibler model links *D* to a number of free water molecules *n* and a number of bound water molecules *N* as a function of time *t* (Equation (Equation 16)) [45].
(15)$$D=\frac{\pi }{16}\frac{h^{2}}{t}{\left(\frac{M(t)}{M_{sat}}\right)}^{2}$$
(16)$$D\frac{\delta^{2}n}{\delta x^{2}}=\frac{\delta n}{\delta t}+\frac{\delta N}{\delta t}.$$

While $M_{sat}$ and *D* increase with RH in the statistical study, $t_{sat}/h$ evolution with RH is less evident. The high dispersion of this parameter does not allow us to conclude and is not suitable for a simple statistical dispersion study.

#### 3.6.3. Temperature

The temperature $T_{ageing}$ increase in the humid environment leads to accelerated ageing, as shown in the statistical study in Figure 5. The more important diffusion leads to an absorbed water mass stabilisation in a shorter time [33,39,54,55]. *D*, either Fick or Dual, is related to temperature by an Arrhenius law [37,145,146].
(17)$$D=D_{0}\;exp\left(\frac{-E_{a}}{RT}\right).$$

$T_{ageing}$ seems to increase *D* and to decrease $t_{sat}/h$. However, the strong dispersion of the data does not allow any conclusion.

In contrast, $M_{sat}$ does not seem to be affected by the temperature. The non-relationship of saturation mass uptake with ageing temperature is under sorption isotherm laws such as Henry’s. The saturation mass uptake does not depend on the temperature for the vast majority of studies, whether they are epoxy resins, epoxy-based composites, with varying reinforcements and architectures, Fick sorption kinetics or not. At 25 °C, epoxies have a solubility parameter δ close to 22 MPa${}^{1/2}$, which is very far from the water solubility parameter, of 47.8 MPa${}^{1/2}$ [117,147,148]. Water and epoxies are then hardly miscible. However, a higher temperature induces a decrease in the water solubility parameter while that of polymers remains reasonably constant or increases slightly. Substances that were not solvent can then become so. From an overall study point of view, and although there are exceptions, the variations in these two solubility parameter do not seem to be sufficient for the epoxies and the water to be miscible enough to induce a correlation between $M_{sat}$ and $T_{ageing}$. For example, at 100 °C, the water solubility parameter is 44.4 MPa${}^{1/2}$. That of the Epikote 828™/Epikure™ epoxy system increases from 21.9 at 25 °C to 22.5 MPa${}^{1/2}$ at 100 °C [149].

However, as with the relative humidity, to observe the temperature correlations and be able to conclude, it is necessary to use another type of statistical analysis.

The wide dispersion of the data makes it difficult to conclude by simply observing the evolution of the variables in pairs. In addition, the data in the study show great heterogeneity due to a large number of prepolymers, hardeners and reinforcements, the diversity of processing methods, and the different conditioning parameters. All these parameters vary from one individual to another, which leads to other absorption behaviours, although irreversible behaviours have already been ruled out. A simple comparison of the variables in pairs is not sufficient to draw conclusions. To complete the statistical study, principal component analysis (PCA) is used in the rest of this paper.

## 4. Principal Component Analysis

### 4.1. Principal Component Analysis Introduction

The dataset used in this paper is associated with a large number of quantitative variables. Each of these variables can be related to a dimension. When the number of variables exceeds three, it is challenging to visualize a multidimensional space. Principal component analysis (PCA) can then be used. This is a multivariate statistical analysis method used when individuals are described by several quantitative variables, and it is used to determine the relationships between these different variables. PCA is based on the projection of a quantitative dataset belonging to a multidimensional space into several two-dimensional spaces. It simplifies the study by reducing the dimensions, i.e., the variable number, by highlighting principal components, which are linear combinations of the original variables. The *n* individuals are projected into a subspace of dimension *q*, which is the principal components space. To represent a data cloud $S_{i}$ in a reduced space, we use a system of *q* linear combinations $CP_{q}$ and *p* quantitative variables $X^{p}$. These linear combinations $CP_{q}$ are the principal components [150,151].

PCA is particularly effective when the variables are highly correlated. In statistics and probability, studying the correlation between two variables is equivalent to studying the intensity of their links [114,116]. The correlations between variables and principal components can be read from a correlation circle of radius 1 where a vector of a given length represents each variable. The vector end coordinate corresponding to the variable on a principal component makes it possible to quantify its correlation. The closer the vector length is to 1, the better the variable is correlated in the principal component. It is also possible to observe correlations between variables if their vectors are long enough, so they are sufficiently well represented in the two-dimensional space. The correlation circle is interpreted as follows [152,153]:An acute angle (<90°) between the vectors of individual variables indicates a positive correlation between them;A 180° angle between the vectors of individual variables indicates a negative correlation between them;Variables whose vectors are orthogonal are not correlated with each other, and are therefore independent.

Therefore, principal component analysis enables, in addition to quantifying the correlation between a variable and a principal component, suggesting correlations between variables. The objective is to identify the dimensions or principal components along which the data variation is maximal. These data are represented in a system of X-Y coordinates. It is then possible to highlight relationships between variables that cannot be visualized in a space with more than two dimensions. Each individual in the study, characterized by these variables, is represented by a point. All these points form a data cloud described in a two-dimensional space. For the principal component analyses, the 448 individuals from the 90 publications are used. The set of variables is given in Table 8.

### 4.2. Data Standardisation

A large dataset may contain heterogeneous individuals. This is particularly the case for some variables of our study, such as $t_{sat}$ and *D*. It is then essential to center and reduce (i.e., to standardize) the dataset, which allows giving the same importance to the all the variables. The standardization is carried out so that the variables have a standard deviation equal to 1 and a mean value equal to 0 [116].

The PCA function of the package *FactoMineR* from *R* automatically normalises the data as explained above [116,154,155,156]. This function generates various indicators for each dimension or principal component: eigenvalues, variances, and cumulative variances. The eigenvalues can be used to determine the number of dimensions to keep for the study of the dataset. For this purpose, the data being centred-reduced, an eigenvalue of >1 is required. This means that the component concerned represents more variance than the original variable alone [157]. The eigenvalue quantifies the variance explained by each dimension. It is large for the first dimensions and small for the following ones (Table 9). Therefore, the first dimensions correspond to the directions that carry the maximum amount of variation contained in the dataset. PCA significantly reduces the number of principal components and compresses the variance into a smaller number of axes.

Another method used to determine the number of the dimensions to choose from is to consider the graph or “scree plot” of the eigenvalues and stop at the level of the eigenvalue drop-off, beyond which they are relatively small (Figure 6) [150,158].

If the whole dataset is considered, it could be possible to stop at the fourth principal component because its eigenvalue is less than 1. However, this only represents a cumulative variance of 60.7%. Looking at the eigenvalue graph, we do indeed see a drop-off at the fourth principal component, but components with relatively large variance follow it. We can choose to stop at this 4th principal component or to stop at the 6th before the second drop-out, that is to say, a cumulative variance of 92.4%. The latter solution is chosen. In addition, by observing the representation quality of the variables, we note that some of them are adequately represented on the principal components 4 and 6. This is the case for *D* and RH. This large number of different variables explains this large number of components to be studied. There is no universal method to decide how many principal components to choose for PCA. It depends on the dataset, the variances of each component, but also on how well the variables are represented in the different components.

### 4.3. Results

#### 4.3.1. Overview

By using the PCA method on the 448 individuals in this study, we obtain individual projection graphs and correlation circles that provide information on the relationship (or non-relationship) between the variables. For each principal component, an individual graph and a correlation circle are obtained. The first two principal components represent respectively 27.2% and 19.4% of the variance of the individuals, i.e., 46.6%. It is necessary to observe the first 6 components so that 92.4% of the variance of the individuals is represented, which makes it possible to have an overall vision of the latter. The number of variables to be studied is therefore reduced from 8 to 6 principal components. The variables which constitute them are determined by analysis of the “coordinate” of their vector on the component, which is the cosine of the angle formed between the vector and the principal component: a cosine close to 1 is desired to have an accurate representation of the component. The 6 Equations (Equation 18)–(Equation 23) below describe the principal components and their compositions, the coefficients being the coordinates of the variables in the principal component:(18)$$CP_{1}=M_{sat}\times 0.96\;+\;M_{inter}\times 0.95\;+\;RH\times 0.55\;+\;h\times 0.18\;-\;T_{ageing}\times 0.08\;+\;D\times 0.08\;+\;t_{sat}\times 0.04\;+\;t_{inter}\times 0.004$$
(19)$$CP_{2}=t_{inter}\times 0.79\;-\;T_{ageing}\times 0.67\;+\;t_{sat}\times 0.59-D\times 0.34-\;M_{inter}\times 0.05\;-\;M_{sat}\times 0.02\;+\;h\times 0.01\;+\;RH\times 0.01$$
(20)$$CP_{3}=h\times 0.75\;+\;D\times 0.53\;+\;t_{sat}\times 0.35\;+\;T_{ageing}\times 0.29+\;t_{inter}\times 0.19\;-\;M_{inter}\times 0.12\;-\;M_{sat}\times 0.11\;+\;RH\times 0.10$$
(21)$$CP_{4}=\;-\;D\times 0.74\;+\;T_{ageing}\times 0.40\;+\;h\times 0.28\;+\;t_{sat}\times 0.22-\;t_{inter}\times 0.16\;+\;RH\times 0.16\;-\;M_{inter}\times 0.04\;-\;M_{sat}\times 0.02$$
(22)$$CP_{5}=t_{sat}\times 0.54\;-\;h\times 0.53\;+\;RH\times 0.40\;+\;T_{ageing}\times 0.23+\;D\times 0.20\;-\;t_{inter}\times 0.13\;-\;M_{inter}\times 0.10\;-\;M_{sat}\times 0.05$$
(23)$$CP_{6}=-RH\times 0.70\;+\;t_{sat}\times 0.33\;+\;T_{ageing}\times 0.27\;+\;M_{sat}\times 0.23\;+\;M_{inter}\times 0.21-\;h\times 0.13\;+\;D\times 0.03\;+\;t_{inter}\times 0.02$$

The first principal component is mainly composed of $M_{sat}$, $M_{inter}$ and to a lower level RH: it is the material moisture absorption. The second main component is dominated by $t_{inter}$, $t_{sat}$ and $-T_{ageing}$. The third component is represented by *h* and *D*: it may illustrate the diffusion through the thickness. The fourth principal component is −D. The fifth principal component is also represented by $t_{sat}$ and *h*, but their coordinates are not very large, so their representation is not very good. Finally, the sixth principal component consists mainly of −RH, whose representation is not good either. For a specific variable, the sum of the squared cosines on all the principal components equals 1. As they add up according to the different principal components, the representation of the squared cosines of the coordinates of the variables makes it possible to quickly visualize which principal components they are associated with, their representation quality, and therefore which correlation circles to study subsequently. However, in the literature, no threshold value of squared cosines or coordinates is formally declared (Figure 7).

The correlation circles of the PCA, formed on the first 6 principal components, are then studied (Figure 8). The reduction by principal components allows us to limit the study to 6 and observe the variables by groups. It is already known that the variables of the following groups are correlated with each other because they are very well represented on their principal component:$M_{sat}$ and $M_{inter}$ are correlated with each other ($CP_{1}$);$t_{sat}$ and $t_{inter}$ are correlated with each other, and are anti-correlated with $T_{ageing}$,which means they decrease when $T_{ageing}$ increases ($CP_{2}$);*h* and *D* are correlated with each other ($CP_{3}$);

The vectors’ placement representing the variables on the correlation circles is analyzed by taking into account their length. Figure 8 comprises the correlation circles of the 1–2, 1–3, 1–6 and 2–3 planes, which allow observation of all the analyses made in the following. The other circles, which do not provide more information, are not shown for the sake of clarity. RH, which was moderately well represented on principal component 1, is very well represented on component 6. The observation of the 1–6 plane confirms the correlation of $M_{sat}$, $M_{inter}$ and RH. On the other hand, $M_{sat}$ and $M_{inter}$ are never correlated with the groups $t_{sat}$, $t_{inter}$ and $T_{ageing}$ or *D* and *h*, so their evolution is independent. A positive correlation is observed between *D* and $T_{ageing}$ on the 2–3 plane, which is in agreement with the literature where Arrhenius laws are established between these two variables. Still on the 2–3 plane, *h*, $t_{sat}$ and $t_{inter}$ are correlated. The greater the *h*, the longer it takes to reach saturation time. Although $t_{sat}$ and $t_{inter}$ are anti-correlated with $T_{ageing}$, it is not possible to conclude on the correlation between the $t_{sat}\;and\;t_{inter}$ group and *D*, their vectors not being of sufficient size. These observations are rather coherent with the conclusions made in the literature.

The points disposition on the individual graph is observed according to the diffusion behaviour type (Figure 9). Whether they follow a Fick law or a diffusion two-step derivative, most individuals are mixed in the graph centre. Nevertheless, there are only two-step specimens that extend along principal component 1, which represents $M_{sat}$, $M_{inter}$ and RH. Furthermore, many samples in first part of the axis are Fick. Therefore, the strongest absorptions are associated with two-step materials, while the weakest are obtained with Fick materials, as we have observed with the box plots analysis. Points corresponding to the two types of behaviour studied stretch along component 2, which represents $t_{sat}$, $t_{inter}$ and $-T_{ageing}$. Their intermediate and saturation times are therefore greater. Finally, the points that extend over component 3, corresponding to the thickness and the diffusivity, are associated with two-step diffusion materials. This material type, therefore, achieves the highest diffusivity.

The graphs of individuals along the principal components are now studied with respect to the qualitative variables presented previously. The objective is to highlight groupings according to one or more qualitative variables. However, the sorting according to the type of prepolymer or hardener does not clearly show any groups of individuals (Figure 10 and Figure 11). All types of prepolymers and hardeners are mixed in the large cluster of graphs. Biobased epoxies, such as epoxidised soybean oil (ESO), blend into the cluster with typical $M_{sat}$ and $t_{sat}$ values [74]. The points that move away from the main cluster are not due to a difference in prepolymer or hardener.

Nevertheless, according to the reinforcement type, the graph makes it possible to distinguish several groupings on the 1–2 plane (Figure 12). In fact, towards the graph centre, the glass or carbon fibre composites are located. Then come the neat resins and finally the composites with flax, hemp, aramid and regenerated cellulose fibres. This group stretches along the principal component 1, which represents $M_{sat}$, $M_{inter}$ and RH, which confirms the dispersion analysis as well as the literature: organic fibre composites absorb more water than inorganic fibre composites or even neat resins. Some points also stretch along component 2 which represents $t_{sat}$, $t_{inter}$ and $-T_{ageing}$. Their intermediate and saturation times are therefore larger. These points represent carbon fibre composites and pure resins. There are no glass fibre composites.

By making the same observations with the type of conditioning, a cluster representing humid air and a cluster representing water are noticeable, whether distilled, seawater or deionised water (Figure 13). The humid air cluster is located in the left part of the first principal component. The individuals in this cluster, due to their smaller RH, show below-average $M_{sat}$ and $M_{inter}$ values. The water cluster extends along with component 1. High water mass uptakes distinguish several individuals immersed in distilled water. The two clusters are stretched along with components 2 and 3, which involve individuals with very long intermediate and saturation times for low temperatures. However, the individuals immersed in water with longer times are still more important than those under air.

About the manufacturing process, no data clustering is distinguished, except for the infused individuals that extend along with the first principal component (Figure 14).

The current dataset mixes neat resins and composite materials, but also materials with different absorption behaviour. The graph of individuals shows us clusters according to the diffusion behaviour and the presence or not of reinforcement as well as its nature.

For the continuation of the PCA study, the individuals are separated into four classified PCA according to their nature-neat resins or composites-and according to their moisture diffusion behaviour—Fick or Dual (Table 10). The objective is to study more precisely the correlations between variables, to compare the different behaviours, to further group the variables into principal components and to observe if new groupings of individuals are made. This separation of individuals also allows the introduction of variables not studied up to this point of the document. For composite materials, these variables are the fibre volume fraction and the fibre architecture. For Dual behaviour, they are $M_{inter}$, $t_{inter}$, β, γ and *K*, which are parameters linked to the Dual–Fick, and Carter and Kibler models, but have no meaning for the Fick model. As a reminder, $M_{inter}$ and $t_{inter}$ are respectively the values of the mass of water absorbed and the time at the intermediate stage. β is the probability that bound water molecules will be liberated, and γ is the probability that free water molecules will be bound. β and γ, therefore, illustrate the changes in the state of the water molecules that modify the diffusion behaviour of the materials. Finally, *K* is a parameter related to the swelling of the polymer (Table 11).

#### 4.3.2. Classified PCA

On correlation circles, it is possible to distinguish four main groups of variables present in all the four classified PCA (Figure 15). $M_{sat}$, $M_{inter}$ and RH represent moisture absorption. The $T_{ageing}$, *D* and *h* relate to the water diffusion speed. β, γ, *K* and *D* are associated with the water molecule diffusion and their placement. $t_{sat}$, $t_{inter}$ and $v_{f}$ are linked to the saturation time. This allows reducing the variable number. For example, the variables β, γ and *K* of the Carter and Kibler model are related to *D*. We also see the relationships between the different variables, confirming the experimental studies and models reported in the literature.

The classified PCA also showed differences. Depending on the material type (neat resin or composite) and the behaviour (Fick or Dual), further conclusions of correlation, anti-correlation or non-correlation can be made. In contrast, limited variations are observed depending on the nature of the epoxy prepolymer or hardener. In the case of neat resins and Fick composites, *D* depends on $T_{ageing}$, $M_{sat}$ on RH and $t_{sat}$ on $T_{ageing}$. In case of the fibre presence, $t_{sat}$ is also influenced by $v_{f}$. All these parameters also depend on *h*, as Fick diffusion occurs in the material thickness, without chemical interaction.

For Dual diffusion materials, some correlations are modified in comparison with Fick diffusion materials. For neat resins, $M_{sat}$ is dependent on *h* as for Fick materials, while for composites, it becomes independent. Moreover, $t_{sat}$, which is independent of *D* for Fick resins, becomes anti-correlated for Dual behaviour. $M_{sat}$ changes from being correlated with *D* for Fick diffusion behaviour to being uncorrelated for Dual diffusion behaviour.

In neat resins, β, γ and *K* are correlated with *D*, anti-correlated with $M_{sat}$ and uncorrelated with $t_{sat}$. For composites, β, γ and *K* are always correlated with *D* and $T_{ageing}$. $T_{ageing}$ accelerates diffusion. It indirectly accelerates the propagation and the bonding and unbonding processes of water molecules. However, β, γ and *K* become uncorrelated with $M_{sat}$ and anti-correlated with $t_{sat}$. From the graphs of the individuals, these three variables are more important when the material is exposed to humid air rather than immersed, which may suggest that moisture diffusion is more complex in the air than in water. It should also be noted that $v_{f}$ become anti-correlated with $M_{sat}$ and correlated with $t_{sat}$, whereas, for the Fick cases, these two last parameters were not related to the mass uptake. The presence or absence of fibres has a strong impact on diffusion behaviour. Indeed, the increase of $v_{f}$ decreases the mass water uptake. When there are too many fibres, they act as a barrier for the free water molecules, which diffuse in smaller numbers and more slowly. Different interactions may take place between the fibres and the water molecules, as differences are observed in the graphs of individuals representing the type of reinforcement used.

## 5. Discussion

With box plots, natural organic fibres appear to be the most sensitive to water, followed by aramid fibres which are synthetic organic fibres. In the study, composites based on organic fibres have a Dual behaviour. The presence of hydrophilic functional groups on their macromolecular skeleton amplifies the diffusion and moisture absorption, which can be higher than that of neat resins. Inorganic fibres such as glass and carbon have lower absorption percentages due to their impermeability. However, they still impact diffusion as they create differential swellings at the fibre/matrix interface leading to concentration gradients and possible degradations.

Another major difference highlighted by the correlation circles is the anti-correlation between $T_{ageing}$ and $M_{sat}$ in the case of Dual composites. $M_{sat}$ is then influenced by $T_{ageing}$ whereas it should not be in reversible diffusion models. Moreover, β, γ and *K* are parameters associated with the binding of water molecules to macromolecules. They change from correlated to $M_{sat}$ for Dual Resins to uncorrelated for Dual Composites.

It is known that when the behaviour derives from the Fick model, the diffusion is not only due to the propagation of water molecules in the free volumes. Bonding interactions can take place between these water molecules and the material. These changes can lead to chemical degradation such as hydrolysis, chain breaks or oxidation. Hygroscopic swelling, cracking, osmotic damage and loss of admixtures or particles may occur [6,8,103,159,160,161,162,163]. In some cases, which were not studied in this paper, the gravimetry curves enable to observe these degradations, which are expressed by significant mass losses [7,22,105,112]. In other cases, weak irreversible alterations can take place in the material during hygrothermal ageing while having a two-stage diffusion. This is the case for many under-crosslinked industrial epoxy systems, which contain unreacted oxirane groups or polar compounds such as amines. These groups, which are very hydrophilic, become preferential sites for the formation of hydrogen bonds, for the initiation of hydrolysis and for the creation of concentration gradients [10,72]. As well, the sizing used on the fibres, the additives added to the matrix and the organic fibres can be hydrophilic, degrade with water and create osmotic degradation [29,133,160,164]. These irreversible modifications, which are hardly perceptible on the absorption gravimetry curve, can be observed by desorption. The mass variation M(t) does not return to its initial value. Mass losses therefore occur during the absorption phase while the mass gain curve continues to increase slowly. Studying the desorption kinetics is then important to verify the presence of hidden mass losses or to highlight a chemical evolution [12,72,165,166,167,168]. As with the Fick equation, the Carter and Kibler and Dual–Fick equations model purely physical diffusion kinetics. When chemical degradation takes place, these models may no longer be representative of the material absorption behaviour. If irreversible phenomena enter the diffusion kinetics, it is possible that this will change the correlations between the different parameters. Because of $T_{ageing}$ and $M_{sat}$ anti-correlation, it is then conceivable that in a non-negligible number of cases, the behaviour of the individuals in the study is not purely related to physical diffusion. Degradations, although slight, may be present, making the chosen model inaccurate. Although the gravity curves fit well, the associated parameters may no longer represent the mechanisms for which they were chosen, such as beta, gamma and *K*. The consequences of an inappropriate model choice are the premature termination of gravimetric monitoring of materials, the non-detection of degradation phenomena that may appear at longer absorption times, inaccurate estimates of $M_{sat}$ whose true value is masked by mass losses, and the performance of material characterisation at inappropriate $M_{sat}$ [169,170].

## 6. Conclusions

In this paper, statistical tools were used to study how the parameters related to hygrothermal ageing influence each other. For this purpose, data were extracted presenting scientific publications from experimental tests carried out on neat epoxy resins and epoxy-based composites. The analysis started with a scattering study of the three most characteristic variables of hygrothermal ageing: saturation mass uptake $M_{sat}$, saturation time $t_{sat}$ and diffusivity *D*, using box plots and scatter plots. The box plots enable us to differentiate these three parameters as a function of some qualitative parameters, such as pre-polymers type, reinforcement type, or process. The results obtained are in accordance with the observations made in the literature.

On the other hand, parameters such as hardener type or architecture reinforcement are more difficult to differentiate, as they are strongly linked to other parameters. For the quantitative parameters $v_{f}$, RH and $T_{ageing}$, the analyses are also more complex due to the high dispersion of the data coming from very diverse individuals.

These results led to combining the dispersion analysis with another type of statistical analysis: principal component analysis (PCA). This analysis, which centres and reduces the variables when they are highly dispersed, simplifies the study despite the variation of many parameters and allows the deduction of correlations between variables. PCA was then performed on different groups of data, classified according to the material nature (neat resin or composite) and the diffusion behaviour (Fick diffusion, or Dual diffusion), to observe if any differences are noticeable. Some connections between variables, identified in the correlations circles and graphs of individuals of the PCA, have been demonstrated many times in the literature. The study becomes more complex with the Dual diffusion behaviour composites. Some correlations between parameters change. In particular, *D* and $T_{ageing}$, correlated in the global PCA, become independents in the PCA of Dual composites. These changes do not correspond to the reversible diffusion models. It seems that, despite a two-step diffusion curve, without apparent mass losses in gravimetry, the Carter and Kibler or Dual–Fick models and their parameter do not seem to fit all individuals.

However, the study carried out in this paper has some limitations, due to the quality of representation of some variables, especially in the dual sorption composites PCA. Global PCA does not result in a significant reduction in the number of principal components (−2), although the classified PCA results in better reductions (from −2 to −6). PCA does not allow any remarkable differentiation according to the nature of the epoxies and their hardeners, as this type of polymer is hydrophilic and hygrothermal ageing depends on many parameters. In these two cases, the box plots allow a better visualisation of the data.

Other variables can be added to extend the study, such as mechanical properties, but this requires that these data have been reported in the literature, which is not always the case.

## Figures and Tables

**Figure 1 polymers-14-02832-f001:**
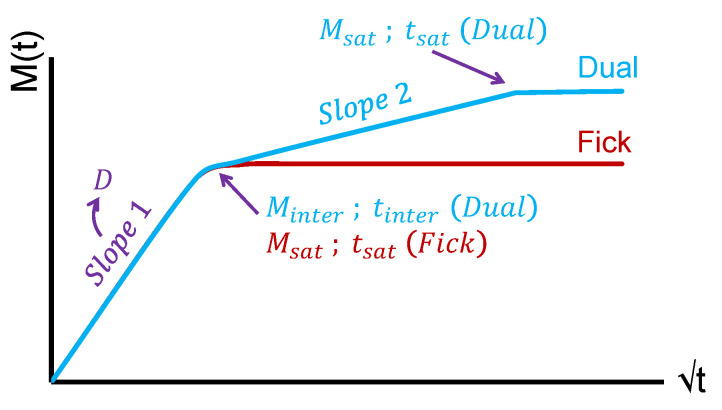
Fick and two-step (Dual) diffusion curves, positions of $M_{sat}$, $t_{sat}$, $M_{inter}$ and $t_{inter}$.

**Figure 2 polymers-14-02832-f002:**
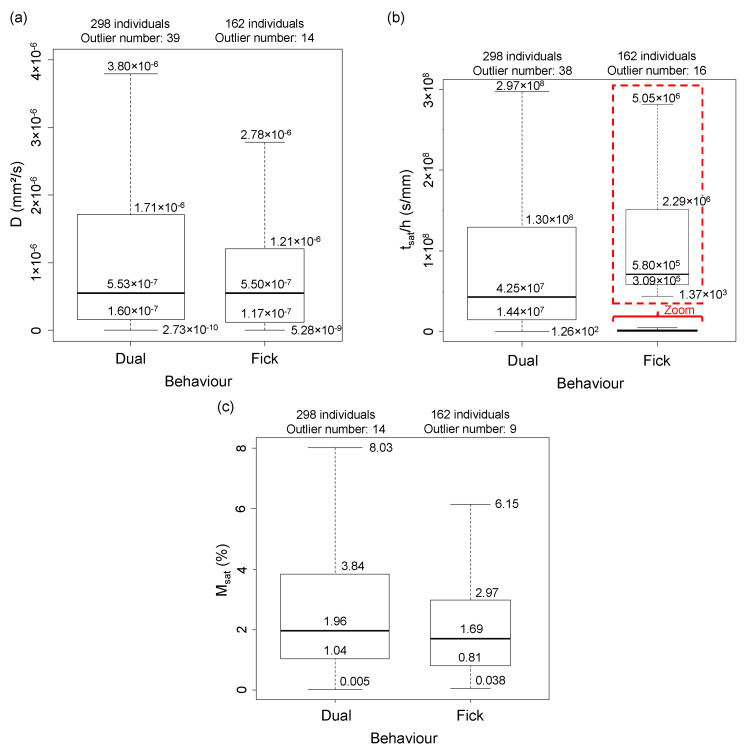
Box plots and characteristic values of (**a**) *D*, (**b**) $t_{sat}/h$ and (**c**) $M_{sat}$ as a function of the diffusion behaviour.

**Figure 3 polymers-14-02832-f003:**
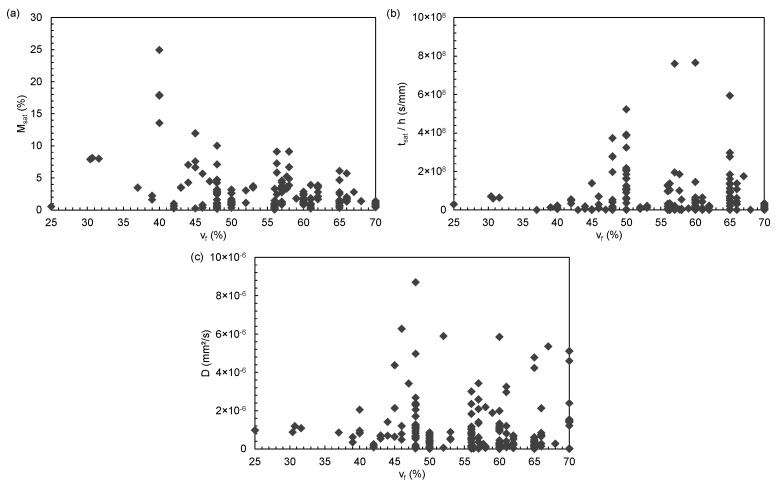
Evolution of (**a**) $M_{sat}$, (**b**) $t_{sat}/h$ and (**c**) *D* as a function of the fibre volume fraction $v_{f}$.

**Figure 4 polymers-14-02832-f004:**
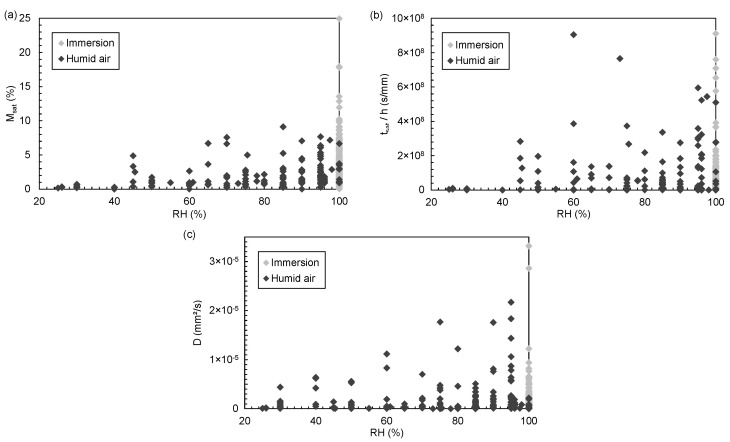
Evolution of (**a**) $M_{sat}$, (**b**) $t_{sat}/h$ and (**c**) *D* as a function of relative humidity RH.

**Figure 5 polymers-14-02832-f005:**
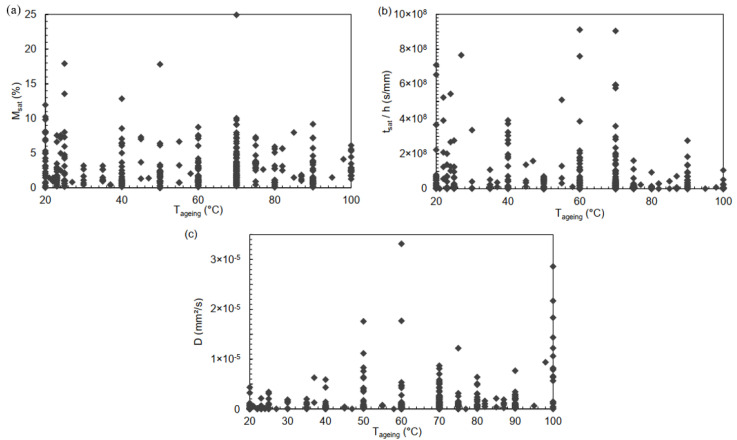
Evolution of (**a**) $M_{sat}$, (**b**) $t_{sat}/h$ et (**c**) *D* as a function of the ageing temperature $T_{ageing}$.

**Figure 6 polymers-14-02832-f006:**
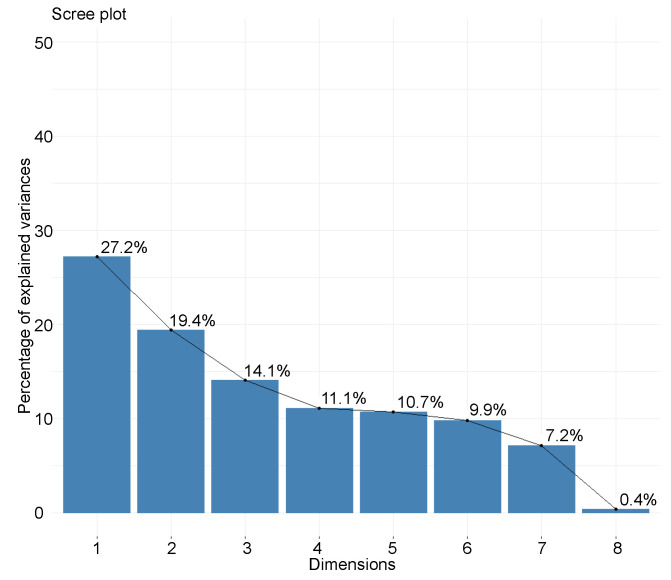
Eigenvalue scree plot.

**Figure 7 polymers-14-02832-f007:**
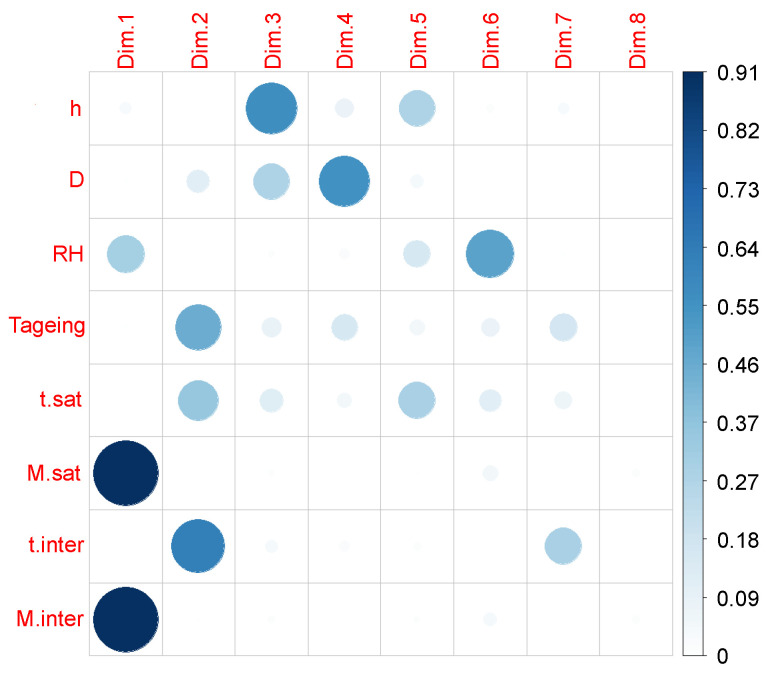
Representation quality of the study variables on the principal components of the overall PCA.

**Figure 8 polymers-14-02832-f008:**
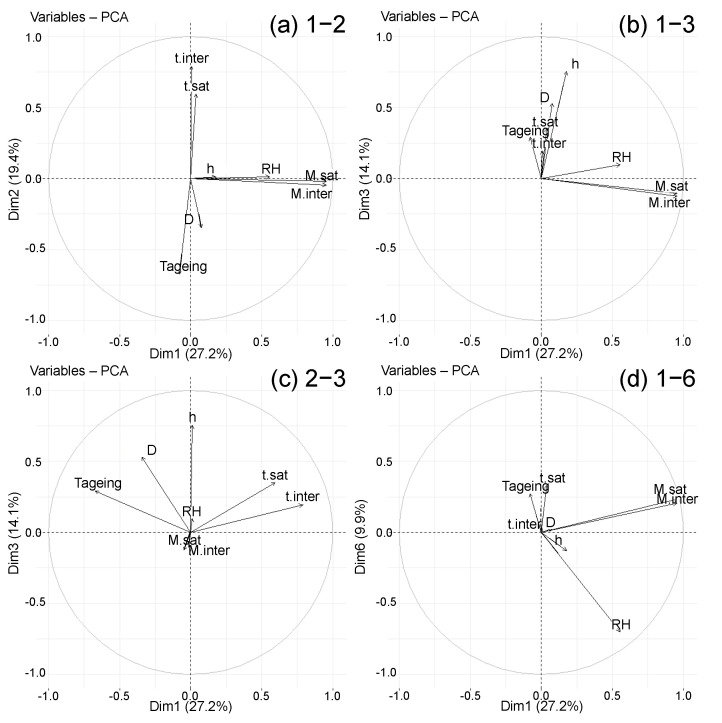
Correlation circles on planes (**a**) 1–2, (**b**) 1–3, (**c**) 2–3 and (**d**) 1–6 of the global PCA.

**Figure 9 polymers-14-02832-f009:**
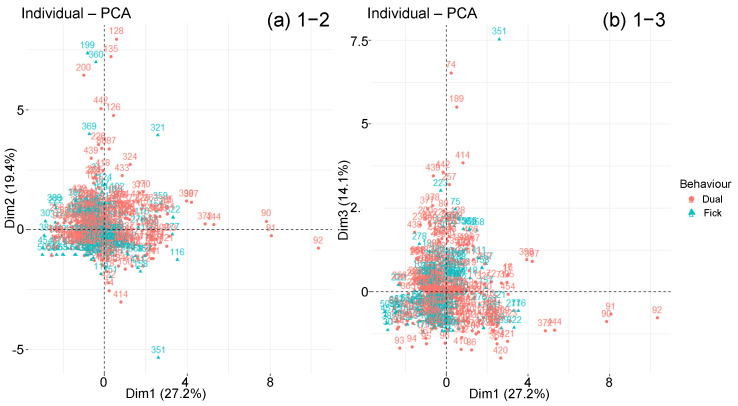
Graphs of individuals on planes (**a**) 1–2 and (**b**) 1–3 of the global PCA, according to diffusion behaviour.

**Figure 10 polymers-14-02832-f010:**
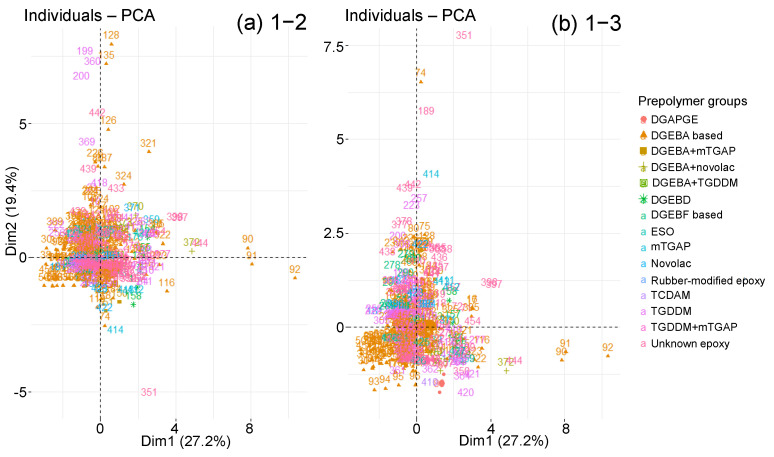
Graphs of individuals on planes (**a**) 1–2 and (**b**) 1–3 of the global PCA, according to prepolymer type.

**Figure 11 polymers-14-02832-f011:**
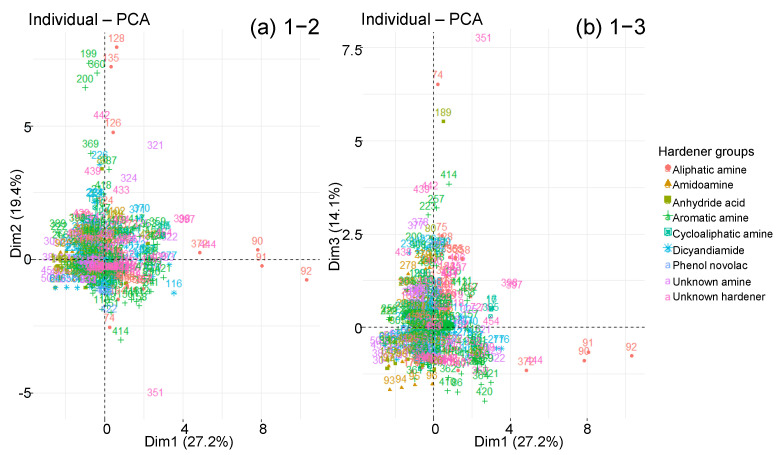
Graphs of individuals on planes (**a**) 1–2 and (**b**) 1–3 of the global PCA, according to hardener type.

**Figure 12 polymers-14-02832-f012:**
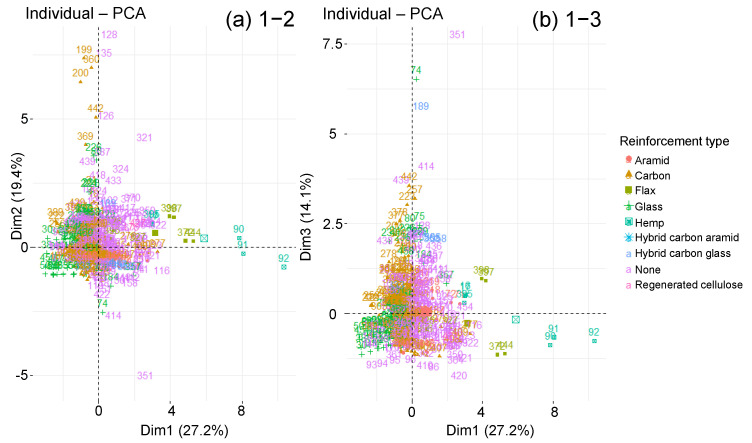
Graphs of individuals on planes (**a**) 1–2 and (**b**) 1–3 of the global PCA, according to reinforcement type.

**Figure 13 polymers-14-02832-f013:**
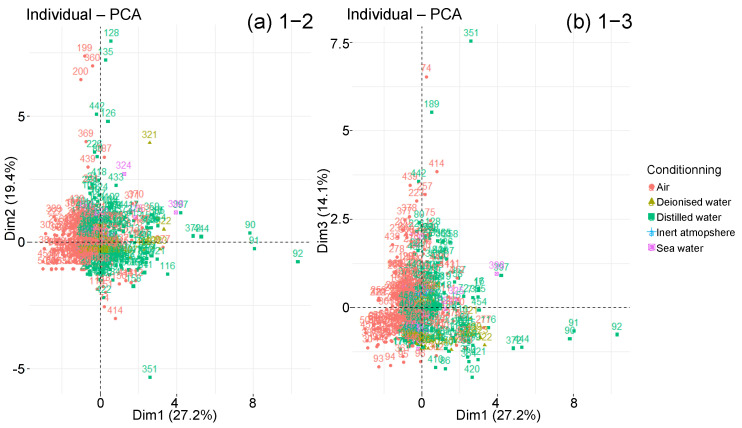
Graphs of individuals on planes (**a**) 1–2 and (**b**) 1–3 of the global PCA, according to conditioning environment.

**Figure 14 polymers-14-02832-f014:**
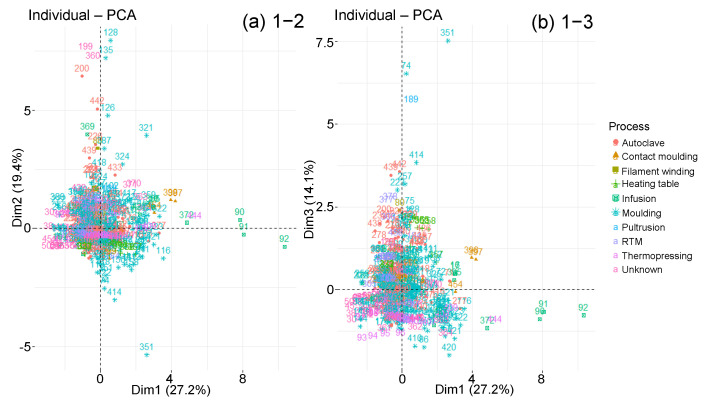
Graphs of individuals on planes (**a**) 1–2 and (**b**) 1–3 of the global PCA, according to manufacturing process.

**Figure 15 polymers-14-02832-f015:**
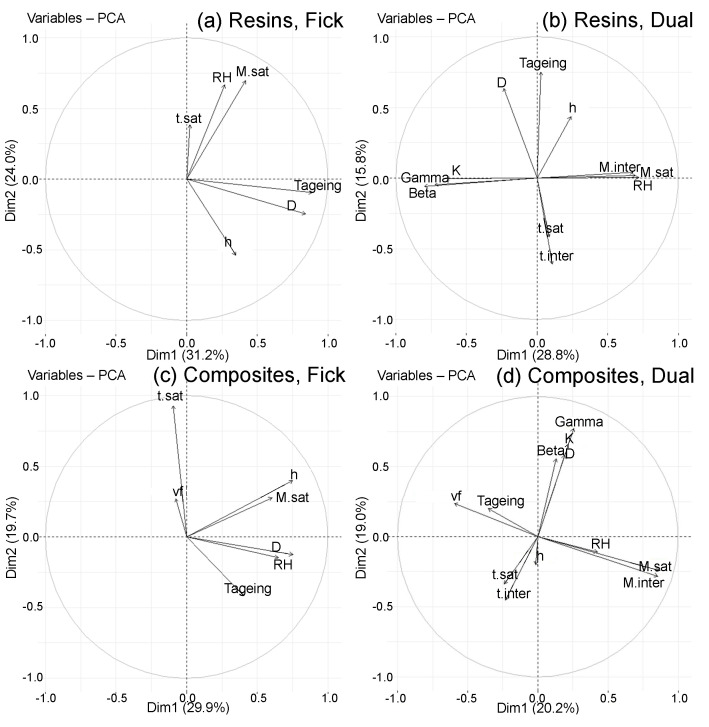
Correlation circles on the 1–2 planes of classified PCA: (**a**) Fick resins, (**b**) Dual resins, (**c**) Fick composites and (**d**) Dual composites. Further planes are given as Appendix A: Fick resins in Appendix A, Fick composites in Appendix A, Dual resins in Appendix A and Dual composites in Appendix A.

**Table 1 polymers-14-02832-t001:** List of variables used for the dispersion study.

	Material Parameters	Ageing Parameters	Diffusion Parameters
Qualitative	Prepolymer type	Conditioning	Diffusion behaviour
	Hardener type		
	Reinforcement type		
	Fibre architecture		
	Process		
Quantitative	$v_{f}$	RH	*D*
	*h*	$T_{ageing}$	$M_{sat}$
			$t_{sat}$

**Table 2 polymers-14-02832-t002:** Medians of $M_{sat}$, $t_{sat}/h$ et *D* as a function of epoxy prepolymer type. List of pre-polymers and entire box plots are available as Appendix A.

Name (Individuals Number)	$M_{\mathrm{sat}}$ (%)	$t_{\mathrm{sat}}/h$ (s/mm $\times \;10^{7}$)	*D* (mm${}^{2}$/s $\times \;10^{-7}$)
DGEBA based (217)	1.61	0.80	5.23
DGEBA + mTGAP (3)	4.44	0.16	106
DGEBA + novolac (3)	6.37	22.30	0.50
DGEBA + TGDDM (2)	4.31	4.96	13.00
DGEBD (5)	6.15	0.02	17.40
DGEBF based (15)	1.05	7.02	3.49
ESO (11)	1.75	0.18	4.67
mTGAP (7)	5.35	0.16	56.61
Novolac (11)	2.26	0.36	15.50
Rubber-modified epoxy (2)	3.08	8.02	0.47
TCDAM (1)	3.84	8.02	0.47
TGDDM (80)	1.50	2.26	4.36
TGDDM + mTGAP (14)	2.02	1.71	6.62
Unknown epoxy (89)	2.64	1.69	9.07

**Table 3 polymers-14-02832-t003:** Medians of $M_{sat}$, $t_{sat}/h$ and *D* as a function of the hardener type. List of hardeners and entire box plots are available as Appendix A.

Name (Individuals Number)	$M_{\mathrm{sat}}$ (%)	$t_{\mathrm{sat}}/h$ (s/mm $\times \;10^{7}$)	*D* (mm${}^{2}$/s $\times \;10^{-7}$)
Aliphatic amine (51)	2.61	1.23	8.51
Aromatic amine (139)	2.63	1.62	5.79
Cycloaliphatic amine (14)	1.81	4.32	3.15
Amidoamine (16)	1.47	0.44	9.18
Dicyandiamide (61)	1.59	5.24	3.46
Anhydride acid (42)	1.01	0.87	4.42
Phenol Novolac (9)	2.10	0.36	5.56
Unknown amine (71)	0.79	0.28	7.33
Unknown hardener (57)	3.20	2.92	8.57

**Table 4 polymers-14-02832-t004:** Medians of $M_{sat}$, $t_{sat}/h$ and *D* as a function of the reinforcement type. “None” refers to a neat resin, without reinforcement. Entire box plots are available as Appendix A.

Name (Individuals Number)	$M_{\mathrm{sat}}$ (%)	$t_{\mathrm{sat}}/h$ (s/mm $\times \;10^{7}$)	*D* (mm${}^{2}$/s $\times \;10^{-7}$)
Aramid (30)	3.54	1.16	6.90
Carbon (127)	1.39	1.53	3.64
Flax (7)	9.82	0.60	14.20
Glass (88)	0.80	0.69	3.63
Hemp (6)	13.0	4.14	9.18
Hybrid carbon aramid (1)	1.62	1.30	3.52
Hybrid carbon glass (7)	3.33	0.04	25.92
Regenerated cellulose (2)	7.10	6.98	13.91
None (192)	2.87	1.59	6.54

**Table 5 polymers-14-02832-t005:** Medians of $M_{sat}$, $t_{sat}/h$ and *D* as a function of architecture reinforcement. Entire box plots are available as Appendix A. UD: one direction, 2D: two directions, 3D: three dimensions.

Name (Individuals Number)	$M_{\mathrm{sat}}$ (%)	$t_{\mathrm{sat}}/h$ (s/mm $\times \;10^{7}$)	*D* (mm${}^{2}$/s $\times \;10^{-7}$)
UD (93)	1.50	1.73	3.49
2D (94)	0.95	1.51	3.99
Balanced (47)	1.80	0.12	8.68
3D (5)	0.99	1.30	6.25
Long Fibre mat (4)	17.87	1.88	9.05
Unknown (2)	2.64	3.21	4.69

**Table 6 polymers-14-02832-t006:** Medians of $M_{sat}$, $t_{sat}/h$ and *D* as a function of the process. Entire box plots are available as Appendix A.

Name (Individuals Number)	$M_{\mathrm{sat}}$ (%)	$t_{\mathrm{sat}}/h$ (s/mm $\times \;10^{7}$)	*D* (mm${}^{2}$/s $\times \;10^{-7}$)
Autoclave (98)	1.52	6.36	3.48
Contact moulding (9)	3.41	2.07	36.60
Filament winding (2)	2.28	66.20	5.52
Heating table (9)	3.33	0.03	21.9
Infusion (28)	1.81	1.48	8.65
Moulding (195)	2.81	1.13	7.00
Pultrusion (1)	2.80	17.5	53.5
RTM (31)	0.89	0.89	7.90
Thermopressing (24)	2.70	2.86	9.48
Unknown (63)	0.94	0.32	1.71

**Table 7 polymers-14-02832-t007:** Medians of $M_{sat}$, $t_{sat}/h$ and *D* as a function of the conditioning environment. Entire box plots are available as Appendix A.

Name (Individuals Number)	$M_{\mathrm{sat}}$ (%)	$t_{\mathrm{sat}}/h$ (s/mm $\times \;10^{7}$)	*D* (mm${}^{2}$/s $\times \;10^{-7}$)
Humid air (215)	1.08	0.57	5.90
Inert atmosphere (2)	2.03	3.88	24.4
Deionised water (20)	3.09	0.89	0.95
Distilled water (203)	3.12	1.99	5.79
Sea water (20)	1.43	2.77	3.50

**Table 8 polymers-14-02832-t008:** List of variables used in the principal component analysis. The 8 qualitative variables are in italics and the 4 quantitative variables in Roman.

Material Parameters	Ageing Parameters	Diffusion Parameters
Prepolymer type	Conditioning	*D*
Hardener type	RH	$M_{sat}$
Reinforcement type	$T_{ageing}$	$t_{sat}$
*h*		$M_{inter}$
Process		$t_{inter}$
		β

**Table 9 polymers-14-02832-t009:** Eigenvalues, variances and cumulative variances of the study dimensions.

Dimension	Eigenvalue	Variance	Cumulative Variance
Dim 1	2.18	27.19	27.19
Dim 2	1.55	19.39	46.58
Dim 3	1.13	14.09	60.68
Dim 4	0.88	11.10	71.78
Dim 5	0.86	10.73	82.52
Dim 6	0.79	9.84	92.37
Dim 7	0.58	7.19	99.56
Dim 8	0.03	0.43	100.00

**Table 10 polymers-14-02832-t010:** Separation of the study individuals into four classified PCA groups.

	All	Resins, Fick	Composites, Fick	Resins, Dual	Composites, Dual
Individual number	448	56	98	147	147
Initial variables number	8	6	7	11	12
Individual variance of each initial variable	12.5%	16.7%	14.3%	9.1%	8.3%
Min. number of PC for an overview	6 (92.4%)	4 (85%)	4 (79.6%)	5 (76.3%)	6 (79.4%)

**Table 11 polymers-14-02832-t011:** List of variables used in the four classified principal component analysis. The 12 qualitative variables are in italics and the 7 quantitative variables in roman.

Material Parameters	Ageing Parameters	Diffusion Parameters
Prepolymer type	Conditioning	*D*
Hardener type	RH	$M_{sat}$
Reinforcement type	$T_{ageing}$	$t_{sat}$
Architecture		$M_{inter}$
Fibre orientation		$t_{inter}$
$v_{f}$		β
*h*		γ
Process		*K*

## Data Availability

The data presented in this study are available on request from the corresponding author.

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
