# Peer review of "Parameters Influencing Moisture Diffusion in Epoxy-Based Materials during Hygrothermal Ageing—A Review by Statistical Analysis"

_polymers, 2022, doi:10.3390/polym14142832_

Round 1
Reviewer 1 Report
Dear Authors,
Please find attached my comments regarding the article sent for review.
Kind regards,
Reviewer

Reviewer 2 Report
At the end of the introduction section, pls summarize main outcomes of previous studies on the subject, highlight clearly research gaps, research significance, and then list the main objectives of the study in bullets.
Figure 2 includes three subfigures. Please provide letters and titles for the subfigures. Discuss each subfigure separately in the text.
Line 180: check grammar.
Figure 3 includes three subfigures. Please provide letters and titles for the subfigures. Discuss each subfigure separately in the text.
Figure 4 includes three subfigures. Please provide letters and titles for the subfigures. Discuss each subfigure separately in the text.
Figure 5 includes three subfigures. Please provide letters and titles for the subfigures. Discuss each subfigure separately in the text.
Figures 8, 9, 10, 11, 12, 13, 14, 15 include subfigures. Please provide letters and titles for the subfigures. Discuss each subfigure separately in the text.
Provide a separate section for the limitation of the study and another section for possible future work.
The conclusion section is very lengthy. It includes many unnecessary texts. This section should be re-written to focus only on main outcomes of the study in bullet format.
